

-

**1**    **Inversion, Assessment of Stability and Uncertainty of Geoelectric Sounding data**

**2**    **using a New Hybrid Meta-heuristic algorithm and Posterior Probability Density**

**3**    **Function Approach**

**4**    Kuldeep Sarkar, Upendra K. Singh*

**5**    Department of Applied Geophysics, IIT (ISM), Dhanbad 826004, Jharkhand, India

**6**    *Correspondence: upendra@iitism.ac.in

**7**    **ABSTRACT**

**8**    Estimating a reliable subsurface resistivity structure using conventional techniques is

**9**    challenging due to the nonlinear nature of the inverse problems. The performance of the

**10**    inversion techniques can be pretty ambiguous based on the optimal error. Although

**11**    traditional methods have proven to be quite effective. The impact of the constraints accessible

**12**    from the borehole is examined for further assessment and enhance the algorithm's

**13**    effectiveness. The vPSOGWO is a new approach based on model search space without any

**14**    prior information. This new strategy describes the hybridization of the particle swarm

**15**    optimizer (PSO) with the grey wolf optimizer (GWO). To understand the efficiency and

**16**    novelty of the algorithm, it has been validated on two different kinds of synthetic resistivity

**17**    data with various sets of noise and finally applied on three field datasets of different

**18**    geological terrains. The analyzed results suggest that the subsurface resistivity model shows

**19**    considerable uncertainty. Thus, it is superior to examine the histograms and posterior

**20**    probability density functions (PDF) of such solutions for exemplifying the global solution.

**21**    PDF with 68.27% CI selects a region with a higher probability. Therefore, the inverted

**22**    models are used to estimate the mean global solution and the most negligible uncertainties,

**23**    where the mean global solution represents the best solution. Our vPSOGWO inverted

**24**    outcomes have been proven to be more accurate than classic PSO, GWO and state-of-art



-

variant of classic approaches. As a results, this novel method plays a vital role in DC data
inversion effectively.
*Keywords*: vPSOGWO, Uncertainty, Stability, Inversion, Resistivity data.
**1. INTRODUCTION**
The vertical electrical resistivity sounding (VES) method is an economical and simple
method due to a wide application such as hydrogeological, groundwater, minerals,
geothermal, hydrocarbon, engineering, environmental fields, etc. (Sen et al., 1993, Sharma,
2012, Panda et al., 2018), which have been used for determining the layered parameters. The
VES data interpretation is challenging due to its unstable, nonunique solution and algorithm
sensitivity (Narayan et al., 1994, Oldenburg and Li, 1994, Singh et al., 2005, 2013).
Therefore, many researchers have developed several inversion algorithms to improve the
accuracy, stability and reduce the uncertainty of the solutions. These inversion techniques are
grouped into local and global optimization techniques. In the local inversion techniques, a
logical initial guess is required to get the solution. The researchers have led to think about
alternative methods, where a broad range of parameters can be established. Many researchers
have developed various metaheuristic optimization algorithms to solve various real-world
problems. These algorithms inspired from the natural phenomenon include Ant Colony
(Colorni et al., 1991), Bat algorithm (Yang, 2010), Biogeographically based Optimization
(Simon, 2008), Differential Evolution (Storn and Price, 1997), Firefly algorithm (Yang,
2010), Genetic Algorithm (Whitley, 1994; Mitchell, 1996), Gravitational Search Algorithm
(Rashedi et al., 2009), Grey Wolves Optimizer (Mirjalili et al., 2014), Particle Swarm
Optimization (Kennedy and Eberhart, 1995), etc. These optimization techniques aim to have
an optimum solution and fast convergent rate to obtain global minima. However, unique
characteristics, viz. exploration and exploitation, in global optimization algorithms persist.





-

For example, the Particle Swarm Optimization (PSO) algorithm has very high potential in
exploitation, implies that the algorithm performs well in local search (Senel et al., 2019) but
is inferior in exploration, which means the algorithm has less ability to find out the starting
position near-global minima and because of low exploration characteristics, it gets trapped at
the local minima (Eiben and Schippers, 1998, Mirjalili and Hashim, 2010). So, integrating the
two algorithms with opposite characteristics is the best way to solve the exploration
characteristics and exploitation characteristics, and provide more accurate and reliable
solution than results obtained from an individual's algorithm. Many authors have developed
various hybrid metaheuristic algorithms such as PSOGA for fundamental function analysis,
PSOACO for data mining, PSODE for global optimization using the standard function, and
PSOGSA using the standard function (Esmin et al., 2013; Lai and Mingyi, 2009; Rashedi et
al., 2009).
This study focuses on a variable weight hybrid algorithm that fuses the exploration
ability of Particle Swarm Optimizer (PSO) with the exploration ability of Grey Wolves
Optimizer (GWO), known as vPSOGWO (Şenel et al., 2019). In this algorithm, some
random particles of PSO are replaced by the new ones obtained from GWO. Earlier the
constant weight hybrid technique of PSO and GWO known as HPSOGWO has been used
in different applications by some authors, such as for single area unit commitment
problems (Kamboj, 2015), mathematical problems (Singh and Singh, 2017), and
benchmark functions and real-world issues (Senel et al., 2019). But none of the researchers
have tested the current work in geophysical data inversion to the best of our information.
Thus, the applicability of the vPSOGWO algorithm is demonstrated on synthetic data with
noise, without noise, and various field resistivity sounding data for estimating the
resistivity distribution in a 1D earth's subsurface model. The study also calculate the
posterior probability density functions (PDF) with 68.27% confidence interval and




-

correlation matrix on all accepted models for determining mean global model and
uncertainty. As a result, we analysed and compared the effectiveness of the proposed
algorithms with classical PSO, GWO and state-of-art variant of classic methods. Our
analysis advocates that the vPSOGWO algorithm produces a more accurate and reliable
model with excellent stabilities and the least uncertainty in the model independently, as
well as the ability to successfully resist noise.

**2.   FORWARD MODELLING ALGORITHM**
The forward code is developed, and synthetic resistivity data sets were created using the
kernel function (Koefoed, 1979) with Schlumberger resistivity configuration (*Fig. 1*) from
known parameters such as current electrode spacing, number of geological multilayers of
true resistivity their thickness. The mathematical expression for apparent resistivity is
given as:
$$\rho_a(s, m) = \rho_1 + s^2\rho_1 \int_0^\infty T_1(\lambda, m)\, J_1(\lambda s)d\lambda \qquad (1)$$
where, $J_1$ is the first order Bessel function, $\lambda$ is the integration variables, $s$ is half of the
current electrode spacing, $m$ is the model. $T_n$ is the kernel's resistivity transform, $\rho_k$ is the
resistivity and $t_k$ is the thickness of the k[th] layers.

For each layer, the kernel's resistivity transform $T_k$ has been determined by Pekeris

(1940). The apparent resistivity, $T_k(\lambda)$, is convolution with linear filter theory to compute
as:
$$T_k(\lambda) = \rho_k * (T_{k+1}(\lambda) + \rho_k \tanh(\lambda t_k))/(\rho_k + T_{k+1}(\lambda) \tanh(\lambda t_k)) \qquad (2)$$



-

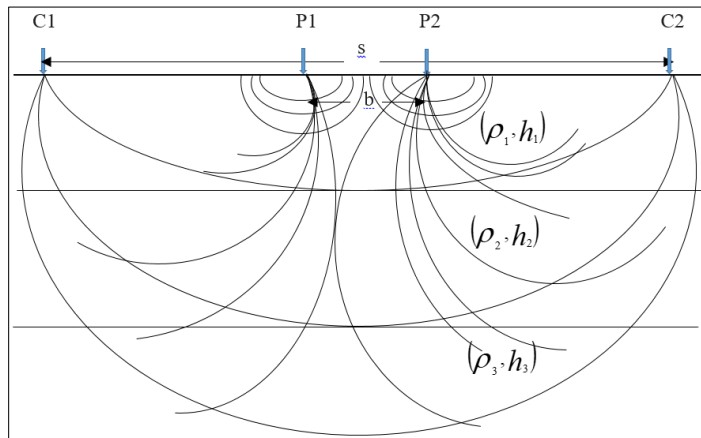


**_Figure 1._** Schlumberger array configuration for three layer case, where C1 and C2, through
which current is injected, are current electrode with spacing s; P1 and P2 are potential
electrodes with spacing b.

## 3. INVERSE MODELLING ALGORITHM

The geophysical inverse problem can be formulated through forward modelling
operator/functional to aim at achieving the geophysical model/solution, which illuminates the
observed data in the best. This operator integrates the geophysical problems and maps
between the observed data $y$ and the solution $x$ as:
$y = f(x)$                                  (3)
Inversion set up finding a model that minimizes cost function/misfit functional that generally
is a degree of the relationship between the N number of observed data $(y_o)$ and the calculated
data $(y_c)$. This misfit functional can be introduced here as a mean-square-error (MSE) and
can be defined as:
$\text{MSE} = \frac{1}{N} \sum_{i=1}^{N} (y_o - y_c)^2$                        (4)





-

### 3.1. Particle swarm optimization

Particle swarm optimization (PSO) is based on the social behavior of animals such as
schooling of fish or flocking of bird (Kennedy and Eberhart in 1995). When the birds go in
search of food, they scattered randomly within the search space before they can determine
the position of food. While searching for food, there is always a bird who is aware of the
position of food. This information they share with others. In this method, each bird is
called as particle which is represented by geophysical solutions/models (i.e., here particle
is resistivity layer parameters). The capability/fitness of each swarm/birds is estimated
between the N number of observed data ($y_o$), which measure the swarm and the food
distance, and the computed data ($\mathbf{y}_c$) which measures the swarm and the estimated position
(resistivity layer parameter/solution) of the prey distance using equation 4.
The best position among particles with information about it are store for each
iteration in memory. The new velocity and position of the population pool are accepted if
its possibility is large, otherwise it is rejected. In that case, the particles are randomly
distributed in the search space in order to escape the local optima. The search continues
until it gains maximum possibility or it reaches the maximum iteration. In global search
space, the position of each particle is updated by the following two mathematical
equations:
$$\vec{\boldsymbol{v}}_i(t+1) = \vec{\boldsymbol{v}}_i(t) + c_1 \times rand\left(\vec{\boldsymbol{x}}_p(t) - \vec{\boldsymbol{x}}_i(t)\right) + c_2 \times rand \times \left(\vec{\boldsymbol{x}}_g - \vec{\boldsymbol{x}}_i(t)\right) \quad (5)$$
$$\vec{\boldsymbol{x}}_i(t+1) = \vec{\boldsymbol{x}}_i(t) + \vec{\boldsymbol{v}}_i(t+1) \quad (6)$$
Here, $\vec{\boldsymbol{v}}_i$ represent the velocity of the $i^{th}$ particle with position $\vec{\boldsymbol{x}}_i$, $\vec{\boldsymbol{x}}_p$ is the best
position obtained by the $i^{th}$ particle, $\vec{\boldsymbol{x}}_g$ is the best position, $t$ is the number of the iteration,
$i$ represents the number of the model ($i = 1, 2, 3, ..., N$), $rand$ represent the random values
with range [0,1], and the coefficient $c_1$ and $c_2$ represent the optimization parameter. The

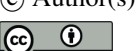



-

disadvantage of PSO algorithm is that, while directing particles to random positions, it has
small possibility to escape the local minima.

**3.2 Grey wolf optimization**

Grey wolf optimization (GWO) algorithm mimics the leadership hierarchy and hunting
mechanics of grey wolves, and used its ability to solve the standard and real-life problems. In
the grey wolf's community, they are divided in four groups: (i) the alpha, (ii) the beta, (iii)
the delta and (iv) the omega, in which alpha, beta and delta are the fittest wolves, who guide
omega towards promising areas of the search space. The alpha is the leader, which generally
makes important and final decision for all the wolves so and represents the fittest solution.
The betas are subordinates that help the alphas in their decision making but they cannot force
them in any decision. They can only order the lower wolves. The beta group takes the order
from alpha group which they reinforce throughout the other group and send back the
feedback to the alpha. All the groups dominate over the omega wolves. The omega group is
an important part during hunting as they play role of the scapegoat and are always allowed to
eat at the end. If a wolf is not the part of alpha, beta or omega group, then they are known as
delta which only summit to alpha and beta groups. In GWO algorithm, the alpha group
represents the best position, i.e., geophysical model/solution. In our case geophysical model
is resistivity layer parameters. The beta and delta groups are consecutive best solutions and
omega group is the best solution that follows always the other groups. The capability/fitness
of each wolf is estimated between the observed data (which measures wolf and prey distance)
and the computed data (which measures the wolf and the estimated position of the prey
distance) using equation 4.





-

Hunting in the grey wolf's community has been divided into three groups: prey
search, encircling the prey, and attacking the prey. The encircling nature of the wolves is
defined by the following equation:
$$d = |c \times (t) - \vec{x}_i(t)| \qquad (7)$$

$$\vec{x}_i(t+1) = \vec{x}_p(t) - a \times d \qquad (8)$$

where, $\vec{x}_p$ is the prey position, $\vec{x}_i$ is the grey wolf's positions, $a$ and $c$ are the vectors
mathematically formulated as:
$$a = a_1 \times (2 \times rand - 1) \qquad (9)$$

$$c = 2 \times rand \qquad (10)$$

Here, $a_1 = 2 \times (1 - t/l)$ which varies from 2 to 0 in decreasing order with
increasing iteration *(t)*, $l$ represent the maximum iteration, and rand is the random
number between [0,1].
The alpha group led the grey wolves' community, in which the beta and the delta
group to search the prey location and the omega groups follow them. The alpha group
wolves gives the best solution, while the second and third best solution is provided by
the beta and the delta group wolves, respectively. Therefore, the rest community wolves
i.e., omega group wolves follows the best solution wolves to obtain best location. This is
mathematical equated by:
$$d_{\alpha,\beta,\delta} = |\vec{c}_{1,2,3} \times \vec{x}_{\alpha,\beta,\delta} - \vec{x}| \qquad (11)$$

The best location/position for alpha, beta and delta wolves in each iteration is
given by $\vec{x}_\alpha$, $\vec{x}_\beta$ and $\vec{x}_\delta$, respectively.
$$\vec{x}_{1,2,3} = |\vec{x}_{\alpha,\beta,\delta} - \vec{a}_{1,2,3} \times \vec{d}_{\alpha,\beta,\delta}| \qquad (12)$$

Here, $\vec{x}_p(t+1)$ describe the updated position of the prey in $(t+1)$ iteration
which is obtained from the mean position of three best wolves in the population, that is,
$$\vec{x}_p(t+1) = (\vec{x}_1 + \vec{x}_2 + \vec{x}_3)/3 \qquad (13)$$



-

The values of $a$ are utilized by wolves which force the search to move away from
the prey. When $a \geq 1$, the hunting is abandoned in order to have a better solution and,
when $a < 1$, the wolves are enforced to attack the prey. In equation 9, $a$ varies between
$[-2a_1,\ 2a_1]$.

**3.3 Hybrid variable weighted PSOGWO (vPSOGWO)**
Despite its usefulness in achieving successful results in real-world problems, it tends to
fall into the local minima, causing the solution to move away from global minima. This
tendency for deteriorating within the local minima is stopped by the exploration ability
of the GWO algorithm. Therefore, the hybrid variable weighted PSOGWO, known as
vPSOGWO that fuses the exploitation potential of PSO with the exploration potential of
GWO to overcome each other's discrepancy with the implementation of varying weight.
Due to the involvement of two distinct variants running together to solve the problem,
this hybrid vPSOGWO is called a co-evolutionary hybrid algorithm. The encircling
behaviour of each wolf is updated by the following equations:
$$\vec{d}_{\alpha,\beta,\delta} = |\vec{c}_{1,2,3} \times \vec{x}_{\alpha,\beta,\delta} - w \times \vec{x}| \qquad (14)$$
$$\text{Here, } w = w_{max} - (w_{max} - w_{min}) \times t/l \qquad (15)$$
Here, $w_{max} = 0.9$, and $w_{min} = 0.2$ are found more appropriate after tuning for our
study.
The best location/position (geophysical model) for alpha, beta and delta wolves in
each iteration is given by $\vec{x}_{\alpha}$, $\vec{x}_{\beta}$ and $\vec{x}_{\delta}$, respectively.
$$\vec{x}_{1,2,3} = |\vec{x}_{\alpha,\beta,\delta} - \vec{a}_{1,2,3} \times \vec{d}_{\alpha,\beta,\delta}| \qquad (16)$$
where,
$$a_{1,2,3} = a_1 * (2 * rand - 1) \qquad (17)$$
$$c_{1,2,3} = 0.5 \ \text{(chosen after tuning)} \qquad (18)$$



-

$a_1 = 2 * (1 - t/l)$          (19)
The updated velocity and position of vPSOGWO are:
$\vec{v}_i(t+1) = w \times \vec{v}_i(t) + c_1 \times rand \times (\vec{x}_1 - \vec{x}_i(t)) + c_2 \times rand \times (\vec{x}_2 - \vec{x}_i(t)) +$

$c_3 \times rand \times (\vec{x}_3 - \vec{x}_i(t))$

(20)

$\vec{x}_i(t+1) = \vec{x}_i(t) + \vec{v}_i(t+1)$          (21)
Here, the value 1.5 for each coefficients $c_1, c_2$, and $c_3$ after tuning the parameters
found more suitable in the present study (Roshan and Singh, 2017).
___________________________________________________________________

**vPSOGWO algorithm**

___________________________________________________________________
*Max_Iter*: maximum iterations set
*Pop_no*: population size
*Para*: Number of parameters
*Fitness=infinite*: already set
*Lb* and *Ub*: set Lower bound (*Lb*) and Upper bound (*Ub*) for different parameters
*Initialize particles randomly*
Procedure
for *l* = 1 to *Max_Iter* do

for *i* = 1 to *Pop_no* do

for *j* = 1 to *Para* do

check the *Lb* and *Ub* for randomly created particles

end

end

for *i* = 1 to *Pop_no* do





-

Calculate the *fitness* form cost function

Update the wolves' fitness and position

end

Update *a1*, *a*, *c*, *w*, using equations (15-17), (13)

for *i* = 1 to *Pop_no* do

for *j* = 1 to *Para* do

Update position of $\vec{x}_1$, $\vec{x}_2$ and $\vec{x}_3$ using equations (14) and (16)

Update best particle velocity and position using equations (20-21)

end

end

end

_______________________________________________________________________

## 4.0  Statistical simulation for global model and uncertainty estimation

The proposed algorithms yield good-fitting models, but the evaluation of a global solution requires numerous techniques. It is noteworthy for selecting the region of solution/model search space, where we find enormous solutions. The methods for selecting the region of model space were selected to envisage the global solution and reduce the uncertainty in the ultimate solution (Mosegaard and Tarantola, 1995; Sen and Stoffa, 1996). Thus, many solutions and associated error estimated were kept in memory for consequent statistical measurements. Therefore, $10^8$ solutions were generated for each algorithm using logarithmic mean square error, and every computed response corresponding to each model fits well with the observed response. However, the model parameters obtained may differ from each other, which lie within the search range in multidimensional space. Hence, the mean model from the model parameters is defined as (Ross, 2009):

$$\hat{m}_i = \frac{1}{M}\sum_{j=1}^{M} m_{i,j} \tag{20}$$



-

where $i = 1$ to the total number of the parameters, $M$ is the total models and $\boldsymbol{m}_{i,j}$.
All algorithms are executed for 10,000 runs with 1000 iterations to obtain the best
model parameters. It is noteworthy to mention that in vPSOGWO, multiple runs are crucial
because 1000 weightage points are laying in between the inertial weights of 0.9 to 0.2, such
that each weightage point yields a fitted model in a run. As a result, 10,000 runs provide
10,000 chances to each weightage point to fetch the best-fitted model.
Therefore, the posterior covariance matrices are defined in the equation (Ross,

2009):

$Cov(\boldsymbol{m}_{i,k}) = \frac{1}{M-1}\sum_{j=1}^{M}(\boldsymbol{m}_{i,j} - \hat{\boldsymbol{m}}_i) \times (\boldsymbol{m}_{k,j} - \hat{\boldsymbol{m}}_k)$          (21)
and posterior correlation matrices are described in the equation:
$Corr(\boldsymbol{m}_{i,k}) = Cov(\boldsymbol{m}_{i,k})/\sqrt{Cov(\boldsymbol{m}_{i,i}) \times Cov(\boldsymbol{m}_{k,k})}$          (22)
where $i$ and $k$ lie between 1 to total number of parameters.
The square-rooted diagonal elements of the covariance matrix define the
uncertainty of the solution, and the correlation matrix gives a rough idea about the relation
between the model parameters. If the parameters don't provide a global solution, then the
apparent resistivity curve corresponding to the mean model will not adequate the observed
value. The posterior correlation matrix corresponding to the indigenous solution will not
yield an actual correlation between the parameters obtained via linear regression. For
further analysis, posterior PDF and histogram are calculated over all accepted models. The
one-dimensional posterior probability density function for various parameters with mean
$\hat{m}_i$ and standard deviation $\sigma_i$ is given as (Ross, 2009):
$p(\boldsymbol{y}_i, \hat{\boldsymbol{m}}_i, \boldsymbol{\sigma}_i) = (1/\boldsymbol{\sigma}_i\sqrt{2\pi}) \times \exp(-(\boldsymbol{y}_i - \hat{\boldsymbol{m}}_i)^2/2\boldsymbol{\sigma}_i^2)$          (23)
where $y$ is the solution/model parameter's output store after 10,000 runs of an algorithm
and $i = 1$ to the number of model parameters.





-

Different techniques are based on the posterior PDF to obtain the global solution.
One of the techniques is to pick the model parameters with the highest probability values.
Another method based on PDF is to normalize (0 to 1) each model parameter by their
respective highest probability values. The best model is considered to have the highest sum
of normalized probability values (Sharma, 2012). Further, the best model can also be
determined by taking the mean of each parameter with probably more significance than the
threshold probability. However, these techniques fail to provide the global model.
Therefore, proceeding with a new approach to the study by introducing a
confidence interval (CI) more significant than 68.27% as a benchmark for all model
parameters. According to the empirical rule, 68.27% of the data lies within the one
standard deviation of the mean (Ross, 2009). Thus, the model parameters below 68.27% CI
are discarded, and the remaining parameters are used for determining the mean solution
and uncertainty. It means that the model represents the global solution with less
uncertainty.

**5.0 Computation information**
The code was developed in MATLAB R2019a in Windows 10 platform having
configuration: Model-HP Z240 Tower Workstation, Processor- Intel Xeon CPU E3-1225
v6 @ 3.30GHz, 32.0 GB RAM, 64-bit operating system (OS). However, Global
optimization is a time-consuming process, as it requires many forwarding calculations to
obtain the best-fitted result.

**6.0 Results and discussion**
The applicability of the new algorithm vPSOGWO, GWO, and PSO has been assessed
inverting several cases of synthetic and field data extracted from different geological



-

terrains (Dixon & Doherty, 1977; Panda et al., 2017). Both synthetic and field data sets
were computed and optimized using the developed algorithms, keeping the ten population
size and 1000 iterations for 10,000 runs, leading each algorithm to analyzed $10^8$ models.
We have discussed the inverted results of algorithms to the application on few examples of
synthetic and field cases:

**6.1 Example 1: Synthetic data- Three-layer case**
Initially, to access the applicability and efficacy of the proposed algorithms, a synthetic
apparent resistivity sounding data measured with Schlumberger array is generated
considering the three-layered earth model sandwich with a high resistive layer of 500.0Ωm
and thickness 150.0 m between two low resistive layers of 8.0Ωm and 5.0Ωm. The
synthetic data is computed in the Matlab environment as shown in *Fig. 2(a)* with the (*)
mark. *Fig. 2* shows (a) the three-layer synthetic data with the best fitted calculated apparent
resistivity curve (> 68.27% PDF) and (b) one-dimensional mean model (> 68.27% PDF)
for true model (black color), vPSOGWO (red color), GWO (blue color) and PSO (green
color).

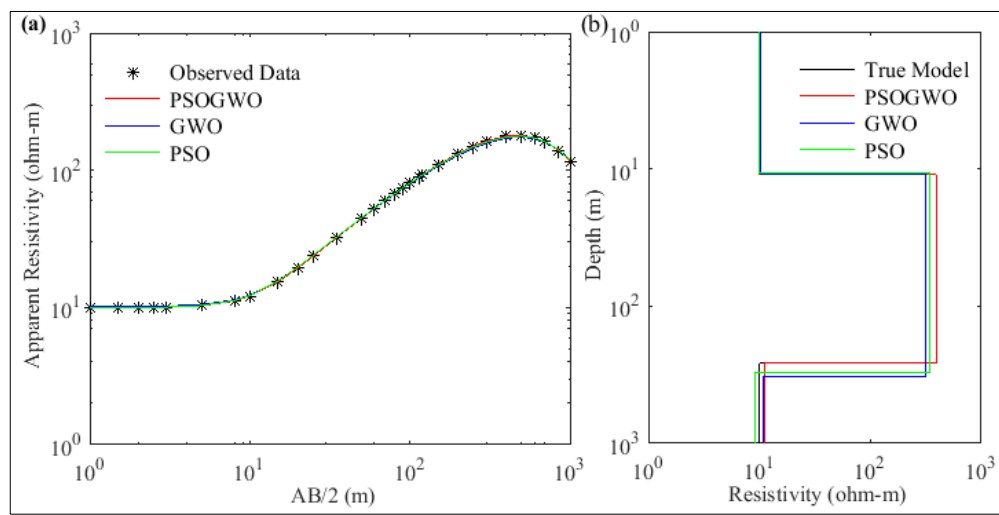




-

**Figure 2.** Three layer synthetic data (a) observed (*) and the best fitted calculated apparent

resistivity curve (> 68.27% PDF); (b) one dimensional mean model (> 68.27% PDF) for true

model (black colour), vPSOGWO (red colour), GWO (blue colour) and PSO (green colour).

The search limit for novel inversions techniques (vPSOGWO, GWO, and PSO) is

carefully chosen, as shown in *Table 1*. Each algorithm, including vPSOGWO, runs 10,000

times to perform statistical analysis and determine the global mean model with the least

uncertainty. *Fig. 3* shows the convergence curve of the resistivity layer parameters using

vPSOGWO. We found no changes seen in the convergence pattern after 590 iterations, and

layer parameters get stable. The convergence curves in terms of error versus iterations for

existed three algorithms are shown in *Fig. 4*. It is observed that vPSOGWO, GWO, and

PSO have converged at 590, 950, and 380 iterations with the mean square error of 1.586e–

8, 5.238e–8, and 5.792e–8, respectively, whereas ridge regression has an error of 0.633.

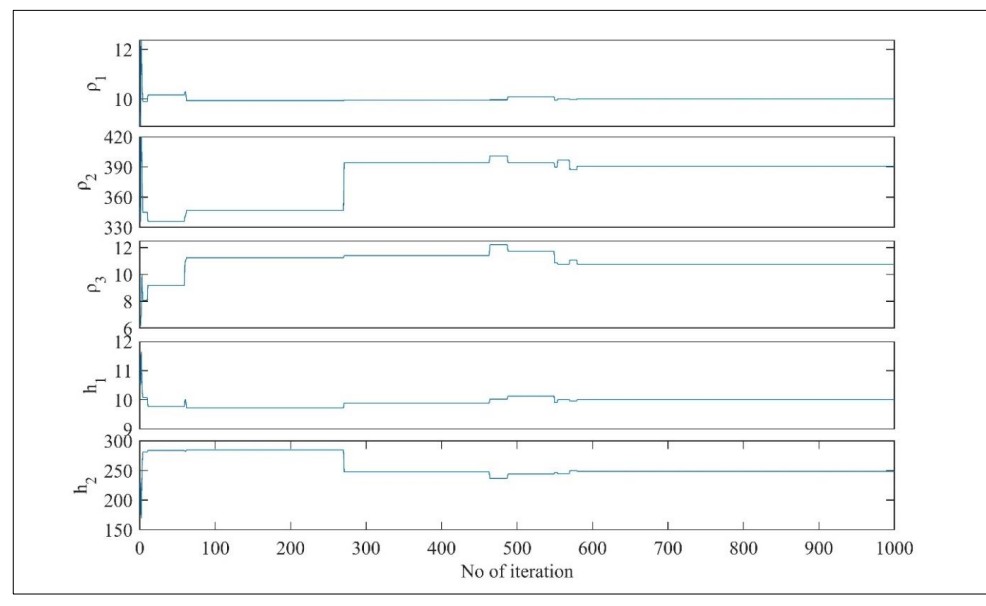

**Figure 3.** Convergence curve for best fitted model parameters for vPSOGWO algorithm.






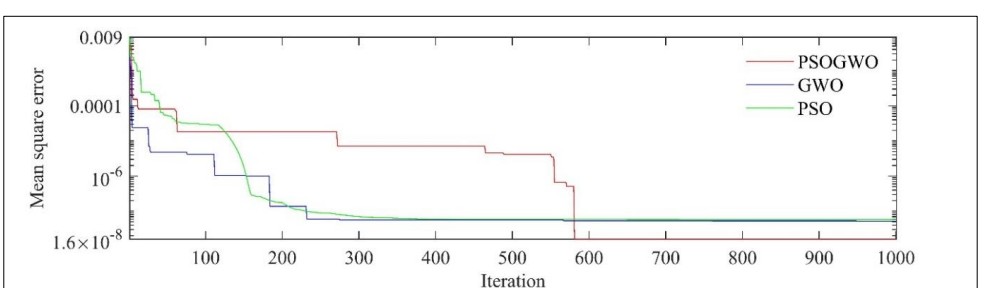


**Figure 4.** Convergent curve known as error versus iteration curve for three layers noiseless

synthetic data.

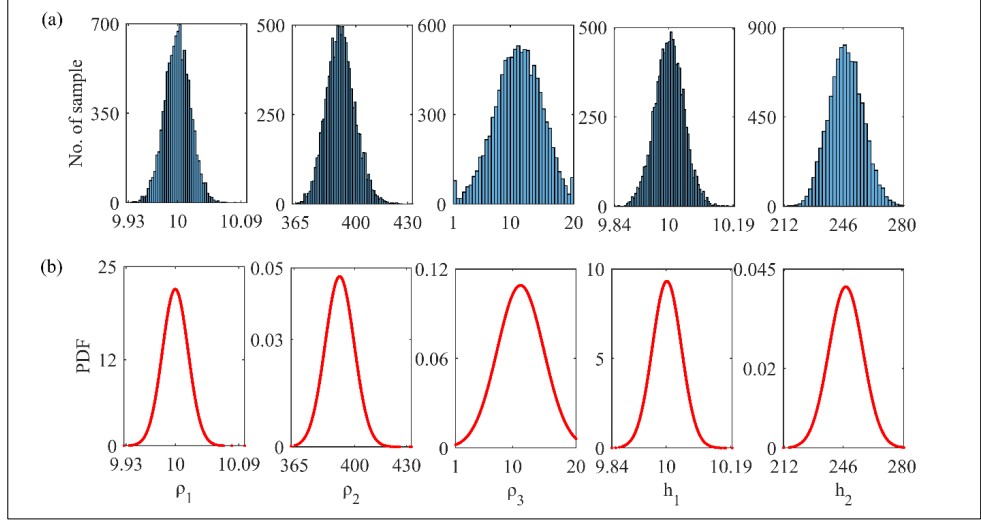


**Figure 5.** (a) Histogram and (b) posterior PDF of all 10,000 solution corresponding to

output of each run for three layer synthetic earth model.


The 10,000 models inverted are used to find out the posterior PDF and histogram
for each parameter. As shown in *Fig.* 5(a), the peak of posterior PDF is roughly close to the
actual model parameter. The histogram is shown in *Fig. 5(b)* suggests that the $\rho_2$ and $h_2$





-

have a broader range. It represents the equivalence problem associated with the resistive
layer as the uncertainty in each algorithm was found to be large considering all the
accepted models. So by selecting the models having posterior PDF greater than 68.27% CI
reduces the uncertainty in the model, increases the resolution of a solution, and helps
estimate the best mean model close to the actual model (*Table 1*). Table 1 shows the model
parameters and uncertainty for proposed algorithms.
***Table 1.*** Optimization mean model result for three layer synthetic resistivity sounding data.

| Model Parameter | True value | Search Range | True model | Mean model (final 10000 solution) | | | Mean model (PDF > 68.27%) | | |
|---|---|---|---|---|---|---|---|---|---|
| | | | | GWO | PSO | vPSOGWO | GWO | PSO | vPSOGWO |
| $\rho 1$ (Ωm) | 10 | 5 – 15 | 10 ± 0.06 | 10.33 ± 0.55 | 10 ± 0.39 | 10 ± 0.02 | 10.15 ± 0.23 | 9.98 ± 0.08 | 10 ± 0.01 |
| $\rho 2$ (Ωm) | 390 | 15 – 500 | 398 ± 8.2 | 324.55 ± 56.71 | 343.10 ± 49.70 | 391.29 ± 8.39 | 319.15 ± 24.02 | 340.90 ± 23.10 | 391.09 ± 3.67 |
| $\rho 3$ (Ωm) | 10 | 1 – 20 | 10 ± 0.05 | 10.50 ± 3.76 | 9.56 ± 7.78 | 11.25 ± 3.66 | 10.71 ± 1.88 | 9.25 ± 2.84 | 11.27 ± 1.70 |
| h1 (m) | 10 | 1 – 20 | 10.1 ± 0.09 | 10.15 ± 0.82 | 9.74 ± 0.56 | 10 ± 0.04 | 9.85 ± 0.33 | 9.72 ± 0.18 | 10 ± 0.02 |
| h2 (m) | 250 | 100 – 500 | 245 ± 4.9 | 314.70 ± 61.46 | 299.55 ± 54.63 | 247.59 ± 9.84 | 312.61 ± 26.91 | 293.21 ± 23.57 | 247.51 ± 3.93 |


***Table 2.*** Correlation matrix using 68.27% PDF limit for three layer synthetic resistivity
sounding data.

| Model Parameter | $\rho 1$ (Ωm) | $\rho 2$ (Ωm) | $\rho 3$ (Ωm) | h1 (m) | h2 (m) |
|---|---|---|---|---|---|
| $\rho 1$ (Ωm) | 1.0000 | –0.0575 | 0.0142 | 0.3820 | 0.0222 |
| $\rho 2$ (Ωm) | | 1.0000 | 0.2585 | 0.6293 | –0.7994 |
| $\rho 3$ (Ωm) | | | 1.0000 | 0.0537 | –0.7678 |
| h1 (m) | | | | 1.0000 | –0.4278 |
| h2 (m) | | | | | 1.0000 |






-

Here, two approaches are used to present the mean solution with its uncertainty
estimation: (i) the mean solution for all accepted best-fitted solutions obtained from 10,000
runs for all three algorithms; and (ii) the mean model calculated from solution with
posterior PDF, which values are greater than 68.27% CI from all accepted solution
parameters.
Here, we observed that the second layer parameters for PSO and GWO are too
diverted from actual values with higher uncertainty due to their inability to balance
exploitation and exploration properties. In contrast, the hybrid vPSOGWO algorithm
provides more accurate results and falls within its uncertainty ranges (*Table 1*). Therefore,
a hybrid algorithm has better exploitation and exploration balancing nature than PSO and
GWO. As shown in *Table 2*, the posterior correlation matrix illustrations that first layer
resistivity reveals a feeble correlation with other associated parameters. Whereas there is a
negative correlation found between $\rho_2$ and $h_2$, both parameters have a trade-off
relationship.

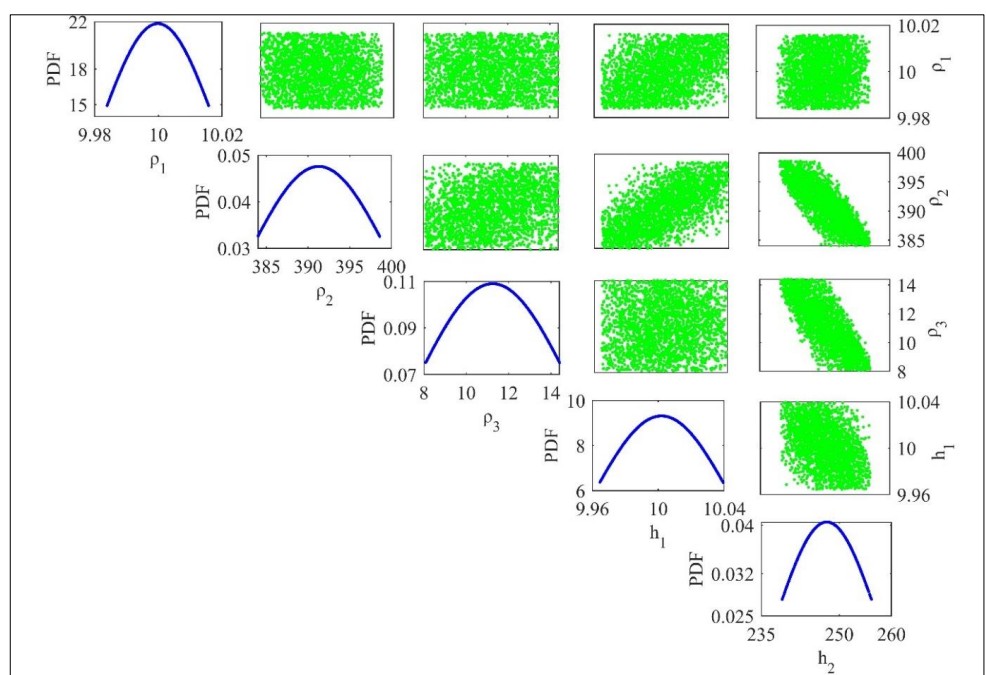






-

***Figure 6.*** Correlation plot between model parameters (off diagonal) and posterior PDF
curve (diagonal) from models having all parameters greater than 68.27% PDF.

In contrast, a positive correlation between $\rho_2$ and $h_1$ is observed (i.e., resistivity of
the second layer increases with increasing the thickness of the first layer and vice versa).
Similarly, it can also be seen between third layer resistivity and second layer thickness but
inverse in nature. *Fig. 6* represents the correlation plot between model parameters (off-
diagonal) with the posterior PDF curve (diagonal) for models greater than 68.27% CI for
all parameters. No significant error differences are found between the observed and
calculated apparent resistivity data for all three algorithms *(Fig. 2(a))*. However, the error
difference in the 1D model and result for 68.27% CI's mean model are presented in *Fig.*
*2(b) and Table 1*, respectively.
***Table 3.*** Stability test for three layer synthetic resistivity sounding data using different
search range.

| Model Parameter | $\rho 1$ ($\Omega$m) | $\rho 2$ ($\Omega$m) | $\rho 3$ ($\Omega$m) | h1 (m) | h2 (m) |
|---|---|---|---|---|---|
| True values | 10 | 390 | 10 | 10 | 250 |
| Search Range | 5 – 30 | 500 – 1000 | 15 – 30 | 1 – 10 | 50 – 90 |
| vPSOGWO | 10 ± 0.02 | 390.44 ± 8 | 10.48 ± 3.60 | 10 ± 0.04 | 249.25 ± 9.93 |
| Search Range | 2.5 - 30 | 7.5 – 750 | 0.1 – 40 | 1 – 40 | 50 - 750 |
| vPSOGWO | 10 ± 0.03 | 398.39 ± 18.01 | 15.93 ± 8.47 | 10.02 ± 0.07 | 237.24 ± 21.98 |
| Search Range | 1 - 60 | 1 – 1000 | 0.01 – 80 | 1 - 80 | 1 - 1000 |
| vPSOGWO | 10 ± 0.03 | 428.11 ± 60.40 | 23.14 ± 13.19 | 10.10 ± 0.15 | 214.86 ± 39.66 |


To check the stability of the parameter, the hybrid algorithm is tested with three
different search spaces, as shown in *Table 3*. Consequently, it estimates the mean model





-

and uncertainty for 100 runs. This Table illuminates using a broader search space suggests
that the result does not divert too much from the actual model. The computations time
required for vPSOGWO, GWO, and PSO are 1.54s, 1.49s, and 1.48s, respectively, for one
run with 30 data points in this example.
***Table 4.*** Optimization mean model result for three layer synthetic resistivity sounding data
with 10% noise.

| Model Parameter | True value | Search Range | Mean model (final 10000 solution) | | | Mean model (PDF > 68.27%) | | |
|---|---|---|---|---|---|---|---|---|
| | | | GWO | PSO | vPSOGWO | GWO | PSO | vPSOGWO |
| $\rho 1$ (Ωm) | 10 | 5 – 15 | 10.37 ± 0.56 | 10.05 ± 0.40 | 10.04 ± 0.02 | 10.21 ± 0.24 | 10.03 ± 0.08 | 10.04 ± 0.01 |
| $\rho 2$ (Ωm) | 390 | 15 – 500 | 323.27 ± 55.51 | 341.58 ± 49.74 | 384.37 ± 7.78 | 317.68 ± 24.39 | 339.42 ± 23 | 384.24 ± 3.41 |
| $\rho 3$ (Ωm) | 10 | 1 – 20 | 10.46 ± 3.79 | 9.57 ± 7.78 | 11.17 ± 3.60 | 10.61 ± 1.94 | 9.35 ± 2.84 | 11.17 ± 1.65 |
| h1 (m) | 10 | 1 – 20 | 10.16 ± 0.83 | 9.75 ± 0.57 | 9.99 ± 0.04 | 9.89 ± 0.35 | 9.74 ± 0.18 | 9.99 ± 0.02 |
| h2 (m) | 250 | 100 – 500 | 314.65 ± 60.48 | 300 ± 54.45 | 251.72 ± 9.59 | 312.96 ± 27.59 | 293.61 ± 23.54 | 251.64 ± 3.82 |


The proposed optimization is also performed using the same synthetic data with

10% Gaussian noise and keeping the search range (*Table 1*). The same procedure is applied
to determine the mean model from all best-fitted solutions and solutions with posterior
PDF greater than 68.27% CI used for parameters of all the solutions (*Table 4*). Although a
10% noise is added, the result obtained from the mean model for posterior PDF of 68.27%
for the hybrid algorithm is not much diverted from actual values. At the same time, the
error was observed that slightly increase 1.309e–5, 1.313e–5, and 1.327e–5 for
vPSOGWO, GWO, and PSO, respectively. *Table 5* depicts the correlation matrix of the
vPSOGWO, which clearly described interdependence by 0.3315 and –0.7879 for the first





-

and second layer's parameters. Similarly, we can also determine the relation between
second layer resistivity and first layer thickness (0.6142), third layer resistivity, and the
second layer thickness (-0.7618). Hence, it shows good agreement with the actual model
values.
**Table 5.** Correlation matrix using 68.27% PDF limit for three layer synthetic resistivity
sounding data with 10% noise.


| Model Parameter | $\rho 1$ (Ωm) | $\rho 2$ (Ωm) | $\rho 3$ (Ωm) | h1 (m) | h2 (m) |
|---|---|---|---|---|---|
| $\rho 1$ (Ωm) | 1.0000 | –0.0816 | –0.0017 | 0.3315 | –0.0552 |
| $\rho 2$ (Ωm) | | 1.0000 | 0.2356 | 0.6142 | –0.7879 |
| $\rho 3$ (Ωm) | | | 1.0000 | 0.0064 | –0.7618 |
| h1 (m) | | | | 1.0000 | –0.3922 |
| h2 (m) | | | | | 1.0000 |






**6.2 Example 2: Synthetic data- Four layers case**
The four-layer earth model having a thin, relatively low resistive (24.0 Ωm) sandwiched
between the two high resistivity layers (840.0 Ωm and 8400.0 Ωm) is considered for
demonstration of the proposed algorithms. *Table 6* illustrates the actual model for synthetic
data, search range, and inverted results. The vPSOGWO, GWO, and PSO converge at
iterations 590, 674, and 750 with associated errors 3.624e–8, 1.370e–8, and 2.097e–7,
respectively as shown in *Fig. 8,* whereas the error estimated using ridge regression method
is 0.383. Instead of higher error in vPSOGWO than GWO, it can also be observed that the
error scale for the vPSOGWO algorithm is narrower than the other two algorithms, which
is an essential factor for determining the mean model (*Fig. 9*). Hence, the mean model is
affected by the error scale, as shown in *Fig. 9.*



-

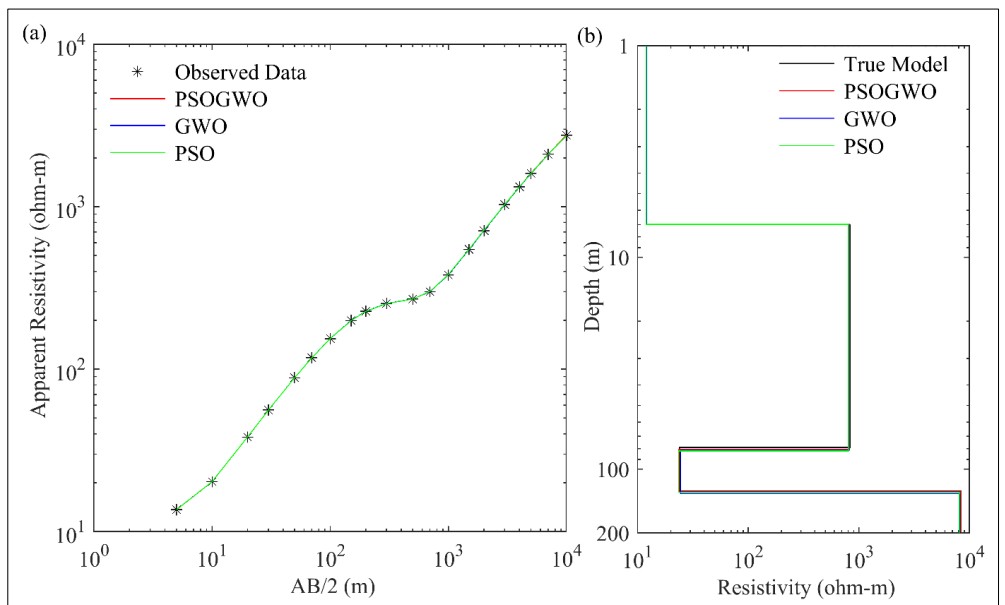


**Figure 7.** Four layer synthetic data: (a) observed (*) and the best fitted calculated apparent

resistivity curve (> 68.27% PDF); (b) one dimensional mean model (> 68.27% PDF) for

true model (black colour), vPSOGWO (red colour), GWO (blue colour) and PSO (green

colour).

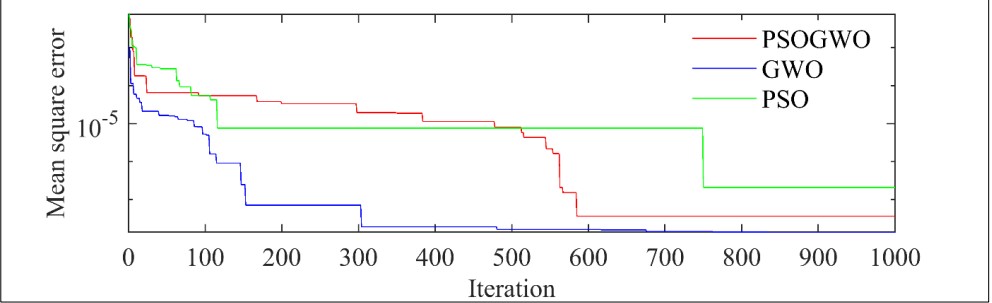


**Figure 8.** Convergent curve known as error versus iteration curve for four layers noiseless

synthetic resistivity sounding data.




-

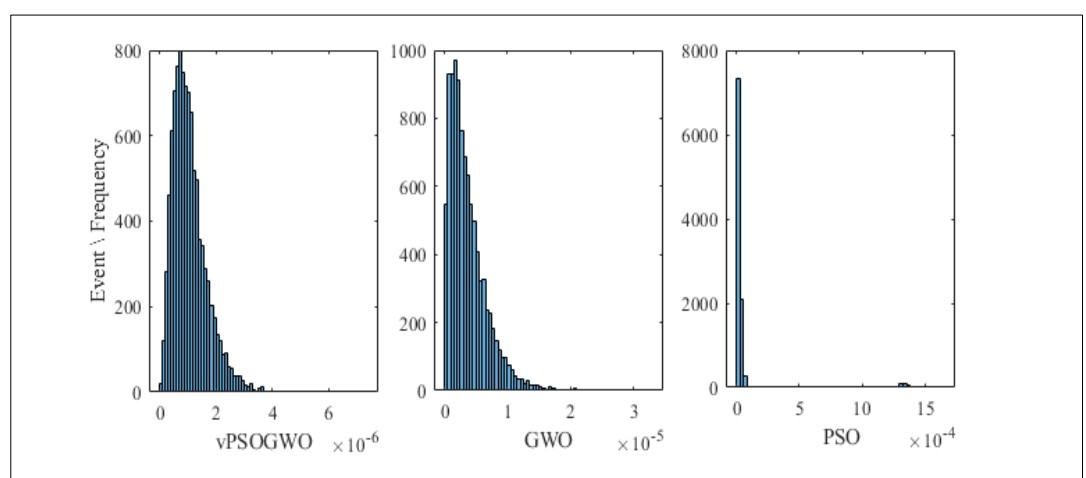


**Figure 9.** Histogram of logarithmic mean square error for vPSOGWO, GWO and PSO

over 10,000 models. The x axis of three histogram represent the misfit error correspond to

10,000 models.

**Table 6.** Optimization mean model result for four layer synthetic resistivity sounding data.

| Model Parameter | True value | Search Range | Ridge regression | Mean model (final 10000 solution) | | | Mean model (PDF > 68.27%) | | |
|---|---|---|---|---|---|---|---|---|---|
| | | | | GWO | PSO | vPSOGWO | GWO | PSO | vPSOGWO |
| $\rho 1$ (Ωm) | 12 | 5 – 30 | 12.1 ± 0.1 | 12.03 ± 0.07 | 12.10 ± 1.05 | 11.99 ± 0.08 | 12.02 ± 0.03 | 12.01 ± 0.39 | 11.99 ± 0.04 |
| $\rho 2$ (Ωm) | 840 | 500 – 1000 | 814 ± 62 | 809.16 ± 28.80 | 802.90 ± 69.13 | 824.36 ± 58.13 | 814.38 ± 10.86 | 803.12 ± 31.07 | 822.71 ± 26.06 |
| $\rho 3$ (Ωm) | 24 | 15 – 30 | 18.2 ± 805 | 24.34 ± 1.30 | 23.78 ± 5.01 | 23.59 ± 3 | 24.50 ± 0.36 | 23.50 ± 1.95 | 23.69 ± 1.41 |
| $\rho 4$ (Ωm) | 8400 | 5000 – 10000 | 7500 ± 3275 | 8151.4 ± 293.68 | 8068.1 ± 614.66 | 8415.50 ± 151.53 | 8150.1 ± 118.05 | 8065.2 ± 301.79 | 8411.9 ± 70.40 |
| h1 (m) | 6 | 1 – 10 | 6 ± 0.07 | 6 ± 0.06 | 6.04 ± 0.68 | 5.99 ± 0.06 | 6 ± 0.03 | 5.99 ± 0.22 | 5.99 ± 0.03 |
| h2 (m) | 72 | 50 – 90 | 74 ± 25.7 | 75.13 ± 2.82 | 75.79 ± 7.36 | 73.99 ± 5.71 | 74.61 ± 0.94 | 75.14 ± 3.20 | 73.77 ± 2.59 |
| h3 (m) | 48 | 30 – 60 | 36 ± 1595 | 48.43 ± 2.71 | 46.98 ± 9.93 | 47.10 ± 5.98 | 48.82 ± 0.88 | 46.46 ± 3.86 | 47.30 ± 2.81 |






-

To reduce uncertainty and increase the resolution of the model, model parameters
containing posterior PDF greater than 68.27% CI are selected. In *Table 6*, the true model
lies within the uncertainty range of hybrid vPSOGWO, whereas GWO and PSO have failed
to keep the true model within its uncertainty range in the second, third, and fourth layer's
parameters. In the case of ridge regression, the uncertainty level of the model parameters is
too high. For example, in the case of the third layer, both resistivity and thickness have
uncertainty approx. 44 times higher than the actual value.

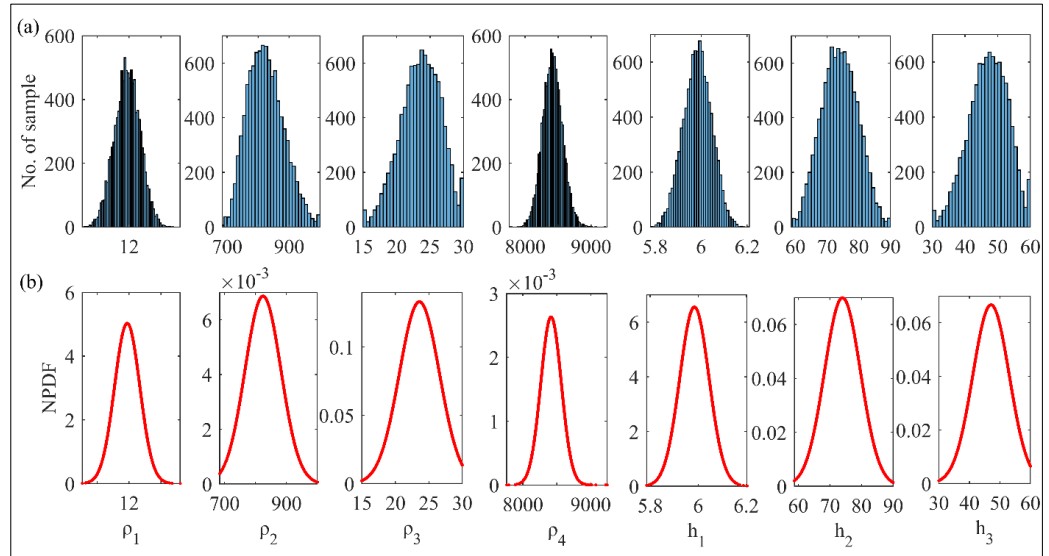


*Figure 10.* (a) Histogram and (b) posterior PDF of all 10,000 solution corresponding to
output of each run for four layer synthetic resistivity sounding data.

The inverted 10,000 models are also computed in this example to find out the
posterior PDF and histogram for each parameter. The peak of posterior PDF is roughly
nearby the actual solution, as shown in histogram *Fig. 10(a) and Fig. 10(b) reveals* the $\rho_2$
and $h_2$ have a broader range that signifies the equivalence problem associated with the
resistive layer. The uncertainty in each algorithm is found to be large considering all the



-

accepted models. However, picking the models with greater posterior PDF than 68.27% CI
reduces the uncertainty in the model, increases the resolution of a solution.

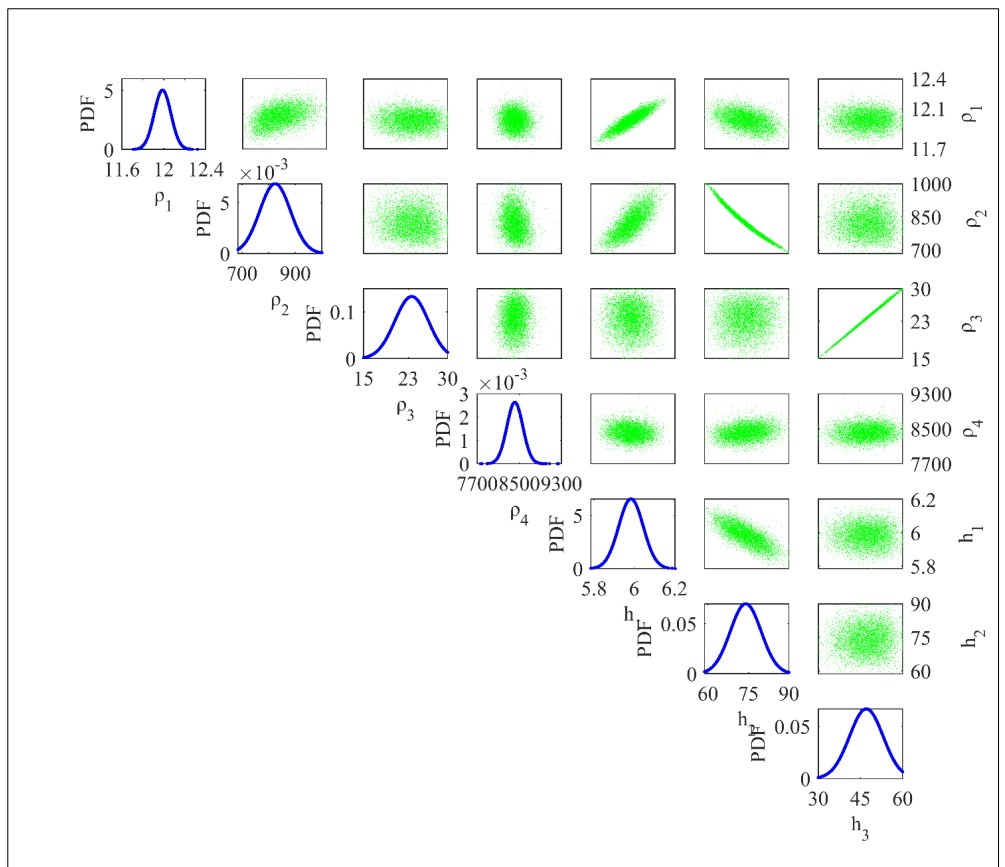


*Figure 11.* Correlation plot between model parameters (off diagonal) and posterior PDF

curve (diagonal) from models having all parameters greater than 68.27% PDF.

The correlation plot between model parameters (off-diagonal) with the posterior

PDF curve (diagonal) for models greater than 68.27% CI for all parameters is shown in
*Fig. 11*. There are also no significant error differences between the computed and observed
apparent resistivity data for all three optimization algorithms.





-

*Table 7.* Correlation matrix using 68.27% PDF limit for four layer synthetic resistivity
sounding data.

| Model Parameter | $\rho 1$ ($\Omega$m) | $\rho 2$ ($\Omega$m) | $\rho 3$ ($\Omega$m) | $\rho 4$ ($\Omega$m) | h1 (m) | h2 (m) | h3 (m) |
|---|---|---|---|---|---|---|---|
| $\rho 1$ ($\Omega$m) | 1.0000 | –0.0359 | –0.0029 | –0.0207 | 0.7383 | 0.0354 | –0.0041 |
| $\rho 2$ ($\Omega$m) | | 1.0000 | –0.0481 | –0.0598 | 0.4667 | –0.9798 | –0.0105 |
| $\rho 3$ ($\Omega$m) | | | 1.0000 | 0.0284 | –0.0188 | 0.0274 | 0.9983 |
| $\rho 4$ ($\Omega$m) | | | | 1.0000 | –0.0183 | 0.0935 | 0.0509 |
| h1 (m) | | | | | 1.0000 | –0.4286 | –0.0036 |
| h2 (m) | | | | | | 1.0000 | –0.0079 |
| h3 (m) | | | | | | | 1.0000 |


The correlation matrix of a four-layer model of synthetic resistivity data is shown in
*Table 7*. It illustrations that the first layer parameters are correlated by a correlation matrix
of 0.7383. A strong negative correlation was found between the second layer parameters (-
0.9798), and the third layer parameters are strongly correlated with each other by a positive
correlation matrix of 0.9983. *Fig. 7(a)* shows the fitness between four-layer synthetic (*)
and computed apparent resistivity data obtained for vPSOGWO, GWO, and PSO. The
difference in fitness curves for all three optimization techniques cannot be determined as
the observed error is significantly negligible. However, the error difference can be
observed in the 1D resistivity-depth models obtained from 68.27% CI's mean model, as
shown in *Fig. 7(b)*. *Table 6* shows the mean model having posterior PDF greater than
68.27% CI for all accepted parameters in the four-layer earth model case. The computation
time for vPSOGWO, GWO, and PSO are 1.94s, 1.84s, and 1.85s (PSO), respectively, for
one run with 27 data points in this example.
The optimization techniques are also executed using the same four-layer model of
synthetic data with 10% Gaussian noise and keeping the search range in *Table 6*. The





-

same procedure is applied to determine the mean model from all the best-fitted models
and models of a posterior PDF greater than 68.27% CI for all model parameters
presented in *Table 8*. Although a 10% noise is added, the result obtained from the mean
model for the posterior PDF of 68.27% for the hybrid algorithm is not much diverted
from actual values. At the same time, the experimental error is 3.831e–4, 3.831e–4, and
3.870e–4 for vPSOGWO, GWO, and PSO, respectively.
***Table 8.*** Optimization mean model result for four layer synthetic resistivity sounding
data with 10% noise.

| Model Parameter | True value | Search Range | Mean model (final 10000 solution) | | | Mean model (PDF > 68.27%) | | |
|---|---|---|---|---|---|---|---|---|
| | | | GWO | PSO | vPSOGWO | GWO | PSO | vPSOGWO |
| $\rho 1$ (Ωm) | 12 | 5 - 30 | 12.25 ± 0.07 | 12.38 ± 1.03 | 12.27 ± 0.09 | 12.24 ± 0.03 | 12.26 ± 0.37 | 12.27 ± 0.04 |
| $\rho 2$ (Ωm) | 840 | 500 – 1000 | 813.70 ± 31.51 | 816.76 ± 66.79 | 901.03 ± 53.95 | 812.08 ± 12.36 | 816.46 ± 29.21 | 899.24 ± 24.66 |
| $\rho 3$ (Ωm) | 24 | 15 - 30 | 24.17 ± 1.36 | 23.51 ± 5.03 | 23.59 ± 2.84 | 24.31 ± 0.42 | 23.28 ± 1.87 | 23.50 ± 1.37 |
| $\rho 4$ (Ωm) | 8400 | 5000 - 10000 | 8070.5 ± 310.96 | 7971.2 ± 596.07 | 8415.50 ± 167.11 | 8082 ± 143.09 | 7973.5 ± 292.28 | 8417 ± 80.27 |
| h1 (m) | 6 | 1 - 10 | 6.15 ± 0.06 | 6.22 ± 0.67 | 5.99 ± 0.06 | 6.15 ± 0.03 | 6.15 ± 0.21 | 6.20 ± 0.03 |
| h2 (m) | 72 | 50 - 90 | 76.80 ± 2.98 | 76.96 ± 6.96 | 73.99 ± 4.59 | 76.72 ± 1.29 | 76.38 ± 3.00 | 69.75 ± 2.10 |
| h3 (m) | 48 | 30 - 60 | 47.35 ± 2.84 | 47.35 ± 10.09 | 47.10 ± 5.85 | 48.75 ± 0.94 | 47.02 ± 3.77 | 48.27 ± 2.83 |


*Table 9* illustrates the correlation matrix of the hybrid algorithm, which clearly
described interdependence by 0.7644, –0.9665, and 0.9980 for the first and second, and
third layers parameters. Similarly, we can also find out the relation between second layer
resistivity and first layer thickness (0.3605) and the resistivity of the fourth layer and





-

thickness of the third layer (0.0549). Hence, it shows good agreement with the actual
model values.
***Table 9.*** Correlation matrix using 68.27% PDF limit for four layer synthetic resistivity
sounding data with 10% noise.

| Model Parameter | $\rho 1$ (Ωm) | $\rho 2$ (Ωm) | $\rho 3$ (Ωm) | $\rho 4$ (Ωm) | h1 (m) | h2 (m) | h3 (m) |
|---|---|---|---|---|---|---|---|
| $\rho 1$ (Ωm) | 1.0000 | 0.0003 | 0.0271 | –0.0948 | 0.7644 | –0.0109 | 0.0251 |
| $\rho 2$ (Ωm) | | 1.0000 | –0.0168 | 0.0327 | 0.3605 | –0.9665 | 0.0153 |
| $\rho 3$ (Ωm) | | | 1.0000 | 0.0260 | 0.0211 | –0.0042 | 0.9980 |
| $\rho 4$ (Ωm) | | | | 1.0000 | –0.0446 | 0.0009 | 0.0549 |
| h1 (m) | | | | | 1.0000 | –0.3180 | 0.0268 |
| h2 (m) | | | | | | 1.0000 | –0.0329 |
| h3 (m) | | | | | | | 1.0000 |



**6.3 Example 3: Field data - Three-layer case**
We have taken one three-layer case of vertical electrical resistivity sounding data measured
with Schlumberger array over Mt. Turner, North Queensland, Australia, interpreted by
Dixon and Doherty (1977, *Fig. 2a*), as shown in *Fig. 12(a)*. After selecting a suitable
search range, three novel algorithms, namely vPSOGWO, GWO, and PSO, are executed to
reconstruct the model interpreted by Dixon and Doherty (1977). The search range and
comparison among proposed algorithms with the previous result (Dixon and Doherty,
1977) are presented in *Table 10*. Our results (for 68.27% CI) are closed to the development
given by Dixon and Doherty (1977). The convergent error for the best-fitted model in
vPSOGWO is 3.681e–4, whereas GWO is 3.697e–4, and PSO is 3.682e–4.





-

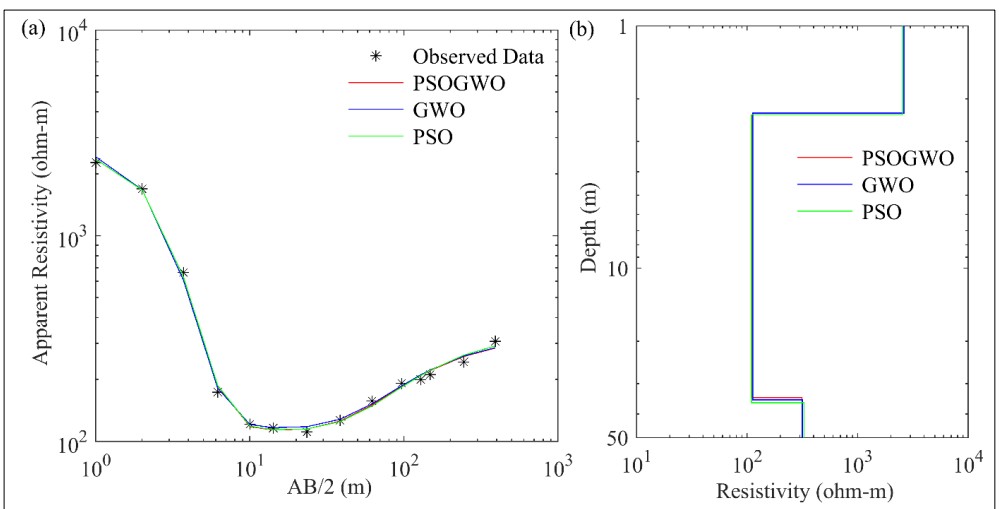

**Figure 12.** Three layer field data over Mt. Turner, North Queenland, Australia: (a) observed (*) and the best fitted calculated apparent resistivity curve (> 68.27% PDF); (b) one dimensional mean model (> 68.27% PDF) for true model (black colour), vPSOGWO (red colour), GWO (blue colour) and PSO (green colour).

**Table 10.** Optimization mean model result for three layer field resistivity sounding data.

| Model Parameter | Search Range | Dixon and Doherty (1977) | Mean model (final 10000 solution) | | | Mean model (PDF > 68.27%) | | |
|---|---|---|---|---|---|---|---|---|
| | | | GWO | PSO | vPSOGWO | GWO | PSO | vPSOGWO |
| $\rho 1$ (Ωm) | 2000 – 3000 | 2500 | 2646.6 ± 246.65 | 2532.3 ± 78.20 | 2536 ± 8.67 | 2619.8 ± 109.70 | 2533.8 ± 34.59 | 2535.9 ± 4.05 |
| $\rho 2$ (Ωm) | 10 – 400 | 100 | 116.01 ± 16.45 | 110.17 ± 3.38 | 109.23 ± 0.29 | 112.55 ± 4.65 | 109.78 ± 1.11 | 109.24 ± 0.13 |
| $\rho 3$ (Ωm) | 200 – 500 | 300 | 318.99 ± 31.67 | 334.01 ± 33.22 | 314.42 ± 1.63 | 315.50 ± 11.96 | 327.15 ± 14.93 | 314.40 ± 0.77 |
| h1 (m) | 0.1 – 3 | 1.42 (approx.) | 1.28 ± 0.13 | 1.33 ± 0.02 | 1.33 ± 0.00 | 1.29 ± 0.05 | 1.33 ± 0.01 | 1.33 ± 0.00 |
| h2 (m) | 20 - 50 | 29.21 (approx.) | 34.02 ± 7.38 | 34.91 ± 6.29 | 31.90 ± 0.31 | 32.66 ± 2.99 | 33.67 ± 2.17 | 31.90 ± 2.17 |



-

**Table 11.** Correlation matrix using 68.27% PDF limit for three layer field resistivity
sounding data.

| Model Parameter | $\rho 1$ (Ωm) | $\rho 2$ (Ωm) | $\rho 3$ (Ωm) | h1 (m) | h2 (m) |
|---|---|---|---|---|---|
| $\rho 1$ (Ωm) | 1.0000 | 0.0046 | –0.0003 | –0.2336 | 0.0086 |
| $\rho 2$ (Ωm) | | 1.0000 | –0.0389 | –0.0897 | 0.3075 |
| $\rho 3$ (Ωm) | | | 1.0000 | 0.0144 | 0.4050 |
| h1 (m) | | | | 1.0000 | –0.0256 |
| h2 (m) | | | | | 1.0000 |


*Table 11* presents the correlation matrix, which shows a negative correlation
between the first layer parameters, and a positive correlation is observed between the
second layer parameters. A positive correlation is also observed between $\rho_3$ and $h_2$, which
maintains the same model data. *Fig. 12(a)* shows the apparent resistivity curve and the 1D
model obtained from the mean model with a 68.27% CI result shown in *Fig. 12(b)*. The
computation time requires for one run in this example with 14 data points is 0.90s
(vPSOGWO), 0.83s (GWO), and 0.78s (PSO), respectively.

**6.4 Example 4: Field data - Five-layer case**
We have selected another field example using a vertical electrical resistivity sounding data
as a five-layer case of earth's subsurface model from Keshiari-Kharagpur near Kharagpur,
West Bengal, India, to determine the aquifer zone (Panda et al., 2018, *Fig. 3*). The area is
covered with different geological units such as laterite, clay, sand, etc., and laterite material
restricts the aquifer's recharge process and most problematic area for groundwater
potential. We inverted this data for a five-layered earth structure parameter using the
vPSOGWO, GWO, and PSO inversion algorithm. The results are shown in *Table 12*
available model, borehole sample, and the search space for vPSOGWO, GWO, and PSO.



-

The computed apparent resistivity curve for all the three algorithms (-) and field data
indicated by the symbol (*) are shown in *Fig. 13(a)*. Their error differences are significant
(*Fig. 13a*, *Table 12*). The inverted 1D layered model using all algorithms obtained from
68.27% CI's mean model is shown in *Fig. 13(b)*. The computations time for vPSOGWO,
GWO, and PSO are 2.55s, 2.43s, and 2.45s, respectively, for one run with 28 data points in
this example.
***Table 12.*** Optimization mean model result for five layer field resistivity sounding data.

| Model Parameter | Search Range | Litho log detail of 100m deep | VES6 (Panda et al., 2017) VFSA | Mean model (final 10000 solution) | | | Mean model (PDF > 68.27%) | | |
|---|---|---|---|---|---|---|---|---|---|
| | | | | GWO | PSO | vPSOGWO | GWO | PSO | vPSOGWO |
| $\rho 1$ (Ωm) | 60 – 120 | - - | 97 ± 5 | 87.97 ± 10.02 | 88.41 ± 13.73 | 78.21 ± 8.28 | 87.44 ± 3.37 | 88.43 ± 5.31 | 77.99 ± 3.17 |
| $\rho 2$ (Ωm) | 10 – 30 | - - | 19 ± 0.2 | 20.38 ± 0.87 | 19.38 ± 1.18 | 19.73 ± 0.17 | 20.43 ± 0.34 | 19.43 ± 0.43 | 19.73 ± 0.06 |
| $\rho 3$ (Ωm) | 80 - 150 | - - | 128 ± 29 | 116.04 ± 10.01 | 118.34 ± 14.41 | 123.24 ± 9.56 | 115.28 ± 3.50 | 117.55 ± 5.67 | 123.01 ± 3.67 |
| $\rho 4$ (Ωm) | 10 - 25 | - - | 60 ± 1 | 16.79 ± 1.31 | 15.27 ± 2.12 | 14.83 ± 0.69 | 16.93 ± 6.49 | 15.35 ± 0.83 | 14.84 ± 0.27 |
| $\rho 5$ (Ωm) | 25 -60 | - - | 40 ± 0.4 | 41.91 ± 2.99 | 44.46 ± 3.60 | 42.83 ± 0.52 | 41.61 ± 1.06 | 44.28 ± 1.35 | 42.67 ± 0.20 |
| h1 (m) | 0.2 – 0.9 | 0.6 (Dry soil) | 0.5 ± 0.1 | 0.54 ± 0.05 | 0.56 ± 0.06 | 0.56 ± 0.02 | 0.53 ± 0.02 | 0.56 ± 0.02 | 0.56 ± 0.01 |
| h2 (m) | 5 – 10 | 7 (Moist soil) | 6.5 ± 0.3 | 7.06 ± 0.56 | 6.35 ± 1.01 | 7.06 ± 0.13 | 7.10 ± 0.21 | 6.36 ± 0.35 | 7.06 ± 0.05 |
| h3 (m) | 6 – 10 | 8 (Compact laterite) | 7.7 ± 2.3 | 8.41 ± 0.72 | 8.78 ± 1.33 | 8.37 ± 0.68 | 8.38 ± 0.26 | 8.77 ± 0.53 | 8.37 ± 0.26 |
| h4 (m) | 40 – 55 | 48 (Soft laterite) | 45.0 ± 5.0 | 51.15 ± 3.57 | 48.34 ± 6.10 | 48.22 ± 3.28 | 51.37 ± 1.37 | 48.60 ± 2.42 | 48.23 ± 1.27 |


* The symbol "- -" in table stand for no information.



-

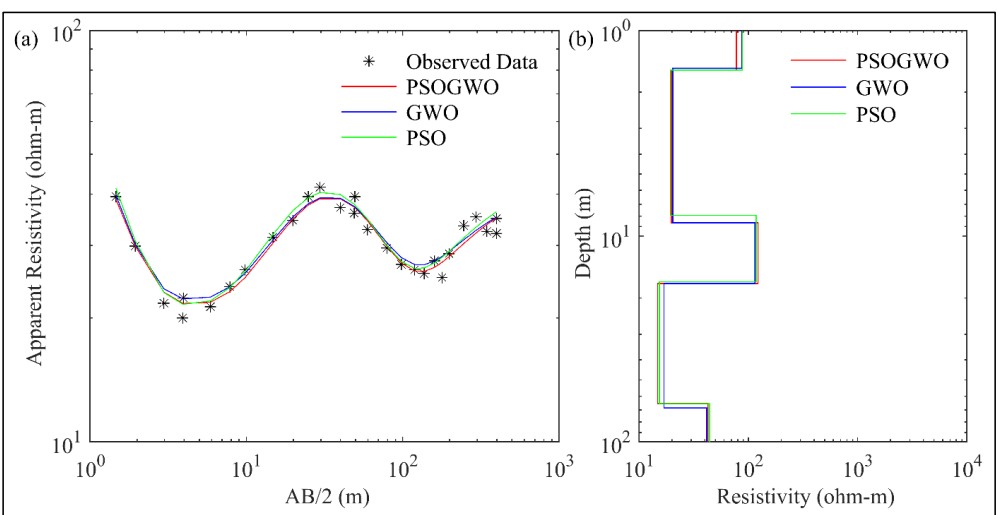


**Figure 13.** Five layer field data: (a) observed (*) and the best fitted calculated apparent

resistivity curve (> 68.27% PDF); (b) one dimensional mean model (> 68.27% PDF) for

true model (black colour), vPSOGWO (red colour), GWO (blue colour) and PSO (green

colour).

**Table 13.** Correlation matrix using 68.27% PDF limit for five layer field resistivity

sounding data.

| Model Parameter | $\rho 1$ (Ωm) | $\rho 2$ (Ωm) | $\rho 3$ (Ωm) | $\rho 4$ (Ωm) | $\rho 5$ (Ωm) | h1 (m) | h2 (m) | h3 (m) | h4 (m) |
|---|---|---|---|---|---|---|---|---|---|
| $\rho 1$ (Ωm) | 1.0000 | 0.8103 | 0.0246 | 0.0164 | 0.1051 | –0.9779 | 0.5888 | –0.0288 | 0.0492 |
| $\rho 2$ (Ωm) | | 1.0000 | 0.1267 | –0.1124 | 0.0684 | –0.8652 | 0.7855 | –0.1035 | –0.0675 |
| $\rho 3$ (Ωm) | | | 1.0000 | –0.1272 | –0.1221 | –0.0390 | 0.6185 | –0.9664 | –0.1169 |
| $\rho 4$ (Ωm) | | | | 1.0000 | 0.4706 | 0.0028 | –0.3107 | –0.0985 | 0.9726 |
| $\rho 5$ (Ωm) | | | | | 1.0000 | –0.1026 | –0.0414 | 0.0449 | 0.6416 |
| h1 (m) | | | | | | 1.0000 | –0.6356 | 0.0392 | –0.0328 |
| h2 (m) | | | | | | | 1.0000 | –0.5463 | –0.2534 |
| h3 (m) | | | | | | | | 1.0000 | –0.0936 |
| h4 (m) | | | | | | | | | 1.0000 |






-

The result obtained from the mean solution of all accepted solutions and solutions
with PDF greater than 68.27% CI aimed at all parameters using the developed techniques
is presented in *Table 12*. The final mean models are comparable with lithological data of
100m deep tube well near VES6. The convergent error for vPSOGWO, GWO, and PSO
are 4.498e–4, 4.541e–4, and 4.566e–4, respectively, whereas the error is 1.7e–2 for VFSA
obtained by Panda et al. (2018). The correlation matrix clarifies a strong correlation
between the parameters of the first layer (–0.9736), the second layer (0.8434), and the third
layer (–0.9907) and a moderate relation between the parameters of the fourth layer
(0.5653). We have noticed a moderate interdependence between $\rho_3$ with $h_2$ and $\rho_5$ with $h_4$,
which follows to retain the same model data shown in *Table 13*.

**6.5 Example 5: Field data - Six layer case**
We again applied the vPSOGWO, GWO, and PSO algorithms to invert the field apparent
resistivity data as a six-layer case study extracted near a borehole from in Apulia, South Italy,
for hydrogeological purposes (Sen et al. 1993). The search range has been taken from Sen et
al. (1993), but the fourth and upper bound thickness of the fifth layers increases by 50 m, as
shown in *Table 14*. The reproduced field data (*) and inverted field data (-) are shown in *Fig.*
*14(a)*. The misfit error obtained is 2.830e–4, 3.243e–4, and 3.133e–4 for vPSOGWO, GWO,
and PSO, respectively, whereas the error using Simulating Annealing (SA) is 0.017 by Sen et
al. (1993). *Table 14* also includes the mean model for 100% and 68.27% CI using proposed
algorithms and previously published literature. It is observed that few parameters obtained
fall within the uncertainty of corresponding parameters of vPSOGWO. The vPSOGWO
inverted results provide higher similarity with the borehole information than the results by
SA (Sen et al., 1993). The interdependence between the layer parameter can be seen from the
correlation matrix as shown in *Table 15*. A strong correlation among parameters of the first


-

layer (0.8211), the second layer (–0.9327), and the third layer (0.9766) has been shown by the
correlation matrix, which is comparable to the correlation matrix that has been presented by
Sen et al. (1933 *Table 13*). A moderate correlation between fourth (–0.5246) and fifth layer
parameters (0.4486) is also observed. It is also to be noticed that there is a sensible relation
between sixth layer resistivity and fifth layer thickness, keeping the same model data.

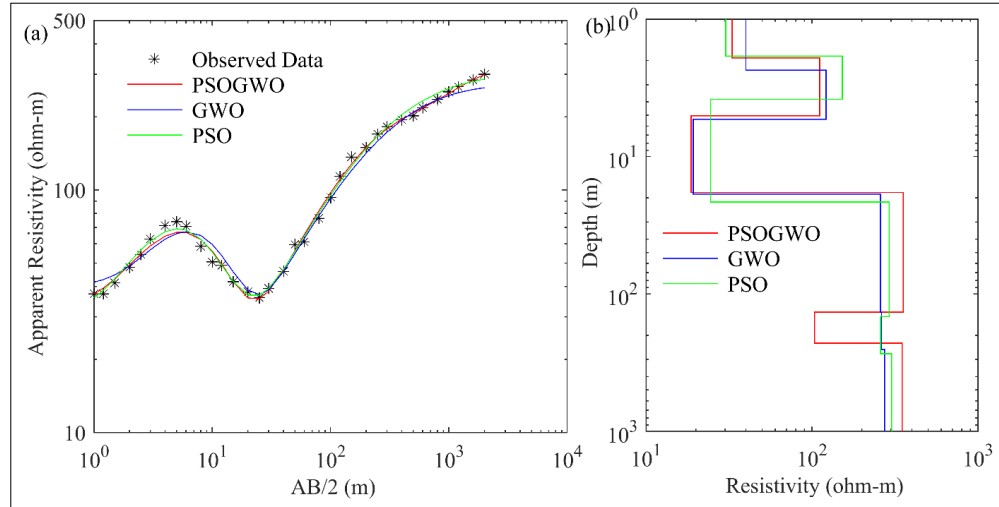


*Figure 14.* Six layer field data over Keshiari-Kharagpur near Kharagpur, India: (a)
observed (*) and the best fitted calculated apparent resistivity curve (> 68.27% PDF); (b)
one dimensional mean model (> 68.27% PDF) for true model (black colour), vPSOGWO
(red colour), GWO (blue colour) and PSO (green colour).
The error differences in computed data with observed data are significant, as shown
in *Fig. 14(a)* and *Table 12*. The inverted 1D layered models obtained from the mean model
of 68.27% CI are shown in *Fig. 14(b)*. The computations time for vPSOGWO, GWO, and
PSO are 3.58s, 3.44s, and 3.45s, respectively, for one run with 28 data points in this
example. The inverted results from vPSOGWO, GWO, and PSO have been shown along
with the borehole data, published result (Sen et al., 1993) in *Table 14*. It can note that the





-

outcomes from the hybrid algorithm satisfy the borehole information provided than the
other algorithms and earlier published results.
*Table 14.* Optimization mean model result for six layer field resistivity sounding data.

| Model Parameter | Search Range | Borehole Detail from Patella, 1975 | Sen et al., 1993 | | Mean model (final 10000 solution) | | | Mean model (PDF > 68.27%) | | |
|---|---|---|---|---|---|---|---|---|---|---|
| | | | | | GWO | PSO | vPSOGWO | GWO | PSO | vPSOGWO |
| $\rho 1$ (Ωm) | 10 - 50 | - - | 37 | 33 ± 4.91 | 36.47 ± 6.23 | 30.00 ± 8.49 | 32.93 ± 1.60 | 40 ± 2.41 | 30.24 ± 2.08 | 33.06 ± 0.57 |
| $\rho 2$ (Ωm) | 50 – 250 | - - | 140 | 240 ± 29.63 | 121.81 ± 29.04 | 158.49 ± 49.17 | 112.32 ± 24.59 | 121.42 ± 11.63 | 152.01 ± 20.51 | 111.25 ± 9.33 |
| $\rho 3$ (Ωm) | 1 – 40 | - - | 17 | 24 ± 1.37 | 19.38 ± 4.58 | 24.14 ± 7.07 | 18.19 ± 3.21 | 19.26 ± 1.85 | 24.49 ± 2.08 | 18.70 ± 1.15 |
| $\rho 4$ (Ωm) | 100 – 600 | - - | 340 | 300 ± 17.5 | 278.55 ± 71.41 | 299.07 ± 53.73 | 355.16 ± 42.70 | 258.02 ± 30.37 | 291.83 ± 23.55 | 354.49 ± 16.04 |
| $\rho 5$ (Ωm) | 30 - 500 | - - | 130 | 120 ± 32.09 | 276.27 ± 80.72 | 265.25 ± 65.06 | 105.80 ± 39.26 | 262.16 ± 33.24 | 259.27 ± 30.44 | 103.67 ± 14.50 |
| $\rho 6$ (Ωm) | 100 – 500 | - - | 300 | 320 ± 8.33 | 286.46 ± 46.72 | 303.76 ± 27.36 | 349.29 ± 20.98 | 273.73 ± 21.91 | 301.75 ± 12.34 | 349.68 ± 7.90 |
| h1 (m) | 0.5 –3 | 1 (Aluvial soil) | 1.3 | 1.1 ± 0.198 | 1.32 ± 0.48 | 0.96 ± 0.66 | 0.91 ± 0.09 | 1.36 ± 0.16 | 0.86 ± 0.10 | 0.92 ± 0.03 |
| h2 (m) | 1 – 8 | 3 (Fine sand) | 2.7 | 1.3 ± 0.252 | 3.17 ± 0.98 | 2.13 ± 1.16 | 3.16 ± 0.47 | 3.01 ± 0.41 | 1.97 ± 0.34 | 3.13 ± 0.18 |
| h3 (m) | 1 – 25 | 12.5 (Calcarenite & sandy clay) | 12 | 17 ± 1.13 | 13.66 ± 3.49 | 17.72 ± 6.03 | 12.93 ± 2.74 | 13.41 ± 1.36 | 17.57 ± 1.94 | 13.26 ± 1.02 |
| h4 (m) | 10 – 200 | 118.5 (Calcareous tufa & limestone) | 120 | 125 ± 8.39 | 117.93 ± 33.89 | 124.38 ± 29.15 | 118.95 ± 30.44 | 117.28 ± 12.31 | 125.08 ± 13.71 | 117.72 ± 11.72 |
| h5 (m) | 10 – 200 | 65 (Water bearing limestone) | 120 | 70 ± 23.15 | 118.79 ± 34.45 | 127.62 ± 29.37 | 93.12 ± 33.99 | 116.89 ± 12.36 | 125.98 ± 13.51 | 92.85 ± 13.03 |






-

*Table 15.* Correlation matrix using 68.27% PDF limit for six layer field resistivity sounding
data.

| Model Parameter | $\rho 1$ ($\Omega$m) | $\rho 2$ ($\Omega$m) | $\rho 3$ ($\Omega$m) | $\rho 4$ ($\Omega$m) | $\rho 5$ ($\Omega$m) | $\rho 6$ ($\Omega$m) | h1 (m) | h2 (m) | h3 (m) | h4 (m) | h5 (m) |
|---|---|---|---|---|---|---|---|---|---|---|---|
| $\rho 1$ ($\Omega$m) | 1.000 | 0.478 | –0.088 | –0.11 | 0.086 | –0.056 | 0.933 | –0.446 | –0.087 | 0.024 | 0.015 |
| $\rho 2$ ($\Omega$m) | | 1.000 | 0.3732 | 0.11 | –0.077 | 0.134 | 0.718 | –0.902 | 0.379 | 0.068 | 0.095 |
| $\rho 3$ ($\Omega$m) | | | 1.000 | 0.54 | –0.388 | 0.392 | 0.005 | –0.661 | 0.988 | 0.021 | 0.186 |
| $\rho 4$ ($\Omega$m) | | | | 1.00 | –0.623 | 0.487 | –0.088 | –0.126 | 0.647 | –0.420 | 0.274 |
| $\rho 5$ ($\Omega$m) | | | | | 1.000 | –0.668 | 0.070 | 0.173 | –0.458 | –0.109 | 0.022 |
| $\rho 6$ ($\Omega$m) | | | | | | 1.000 | –0.027 | –0.223 | 0.449 | 0.324 | 0.528 |
| h1 (m) | | | | | | | 1.000 | –0.655 | 0.006 | 0.044 | 0.033 |
| h2 (m) | | | | | | | | 1.000 | –0.655 | –0.068 | –0.131 |
| h3 (m) | | | | | | | | | 1.000 | –0.033 | 0.217 |
| h4 (m) | | | | | | | | | | 1.000 | –0.014 |
| h5 (m) | | | | | | | | | | | 1.000 |



## 7.0 CONCLUSION

We have evaluated three meta-heuristic algorithms such as PSO, GWO, and vPSOGWO to
realize their efficacy and applicability in the geoelectrical inverse problems, which narrates
the appraisal of 1D resistivity models from geoelectrical resistivity sounding data. The
relevance of these algorithms validated using synthetic and field resistivity sounding data
signifying the kinds of earth's subsurface stratigraphy. An enormous solution 569
(100,000,000 from 10,000 runs) is assessed. Subsequently, the best-fitted solutions are
chosen within a pre-distinct value for statistical measurements. The statistical study
includes posterior PDF with 68.27% CI, a mean solution, posterior solution correlation
matrix, and covariance matrix using search space, was carried out to refine the solutions to
obtain the global mean solution with the least uncertainty. These statistical simulations



-

yield essential information as to the reliability of an inversion algorithm. In general,
conventional techniques can be quite effective in resolving the model in random noise but
can fail in systematic error and inappropriate models. Our investigation with the
application of the developed algorithm, including statistical simulation for different
multilayer resistivity parameters, resulted in a quantitative appraisal of uncertainty in the
derived model parameters. We observed that the output of the hybrid algorithm in terms of
mean model or error might be similar to either PSO or GWO (attributed to the exploration
characteristics of GWO and exploitation characteristics of PSO). The vPSOGWO, GWO,
and PSO algorithms performances have been analyzed based on the uncertainty and
stability and mean model of layered earth structure. We found that the vPSOGWO gives
very closer results than the results inverted from other two algorithms and also
conventional methods which is consistently better than the previously published results,
and correlated well with borehole information.

**CONFLICT OF INTEREST**
There are no conflicts of interest declared by the authors.

**DATA AVAILABILITY STATEMENT**
The data the support the findings of this study will be available on the request from
corresponding authors. All the data taken for study to demonstrate our developed algorithms
are a published/public domain data that obviously written in the manuscript.

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

Methods in Geochemistry and Geophysics: Elsevier Science Ltd. Co., Amsterdam,
14A, pp 276.
Kamboj, V. K., 2015. A novel hybrid PSO–GWO approach for unit commitment Problem.
Neural Computing and Applications 27(6), 1643-1655. https://doi.org/10.1007/s00521-

671 015-1962-4.

Lai, X., Zhang, M., 2009. An efficient ensemble of GA and PSO for real function
optimization. In 2009 2nd IEEE International Conference on Computer Science and
Information Technology IEEE, 651-655. https://doi.org/10.1109/ICCIA.2010.6141614.
Mirjalili, S., Hashim, S. Z. M., 2010. A new hybrid PSOGSA algorithm for function
optimization. In 2010 International conference on computer and information
application IEEE, 374-377. https://doi.org/10.1109/ICCIA.2010.614161.
Mirjalili, S., Mirjalili, S.M., Lewis, A., 2014. Grey wolf optimizer. Advances in
engineering software 69, 46-61. https://doi.org/10.1016/j.advengsoft.2013.12.007.





-

Mosegaard, K., Tarantola, A., 1995. Monte Carlo sampling of solutions to inverse problems.
Journal of Geophysical Research Atmospheres 1001, 12431-12448.
https://doi.org/10.1029/94JB03097.
Mitchell, M., 1998. An introduction to genetic algorithms. A Bradford Book, The MIT Press.
Narayan, S., Dusseault, M. B., Nobes, D. C., 1994. Inversion techniques applied to resistivity
inverse problems. Inverse Problems 10, 669–686.
Oldenburg, D. W., Li, Y., 1994. Inversion of induced polarization data. Geophysics 59,
1327–1341. https://doi.org/10.1190/1.1443692.
Panda, K. P., Sharma, S. P., Jha, M. K., 2018. Mapping lithological variations in a river
basin of West Bengal, India using electrical resistivity survey implications for
artificial recharge. Environmental Earth Sciences 77, 1-10.
https://doi.org/10.1007/s12665-018-7813-8.
Parasnis D. S., 1980. Principles of Applied Geophysics. Fourth ed., New York, Chapman,
and Hall.
Pekeris, C. L., 1940. Direct method of interpretation in resistivity prospecting. Geophysics
5, 31-42. https://doi.org/10.1190/1.1441791.
Rashedi, E., Nezamabadi-Pour, H., Saryazdi, S., 2009. GSA: a gravitational search
algorithm. Information sciences 179, 2232-2248.
https://doi.org/10.1016/j.ins.2009.03.004.
Roshan, R., Singh, U. K., 2017. Inversion of residual gravity anomalies using tuned PSO.
Geoscientific Instrumentation Methods and Data Systems 6, 71-79.
https://doi.org/10.5194/gi-6-71-2017.
Ross, S., 2009. Probability and statistics for engineers and scientists. Elsevier, New
Delhi, 16, 32-33.





-

Sen, M. K., Bhattacharya, B. B., Stoffa, P. L., 1993. Nonlinear inversion of resistivity
sounding data. Geophysics 58, 496-507. https://doi.org/10.1190/1.1443432.
Şenel, F. A., Gökçe, F., Yüksel, A. S., Yiğit, T., 2019. A novel hybrid PSO–GWO
algorithm for optimization problems. Engineering with Computers 35, 1359-1373.
https://doi.org/10.1007/s00366-018-0668-5.
Singh, N., Singh, S. B., 2017. Hybrid algorithm of particle swarm optimization and grey
wolf optimizer for improving convergence performance. Journal of Applied
Mathematics, 1-15. https://doi.org/10.1155/2017/2030489.
Singh, U. K., Tiwari, R.K., Singh, S.B., 2005. One-dimensional inversion of geo-electrical
resistivity sounding data using artificial neural networks—a case study. Computers &
Geosciences 31, 99-108. https://doi.org/10.1016/j.cageo.2004.09.014.
Singh, U. K., Tiwari, R.K., Singh, S.B., 2013. Neural network modeling and prediction of
resistivity structures using VES Schlumberger data over a geothermal area.
Computers & Geosciences 52, 246-257. https://doi.org/10.1016/j.cageo.2012.09.018.
Sharma, S. P., 2012. VFSARES—a very fast simulated annealing FORTRAN program for
interpretation of 1-D DC resistivity sounding data from various electrode
arrays. Computers       &       Geosciences 42,       177-188.
https://doi.org/10.1016/j.cageo.2011.08.029.
Simon, D., 2008. Biogeography-based optimization: IEEE transactions on evolutionary
computation 12, 702-713. https://doi.org/10.1109/TEVC.2008.919004.
Storn, R., Price, K., 1997. Differential evolution–a simple and efficient heuristic for global
optimization over continuous spaces. Journal of global optimization 11, 341-359.
https://doi.org/10.1023/A:1008202821328.
Whitley, D., 1994. A genetic algorithm tutorial, Statistics and Computing 4, 65-85.
https://doi.org/10.1007/BF00175354.



-

Yang, X.S., 2010. A new metaheuristic bat-inspired algorithm. In nature inspired

cooperative strategies for optimization (NICSO 2010), Springer, Berlin, Heidelberg.

65-74. https://doi.org/10.1007/978-3-642-12538-6_6.