# Peer review of "\*Correspondence: upendra@iitism.ac.in"

_Nonlinear Processes in Geophysics, 2022_

## Referee Comment (RC1)

**Comments to Author**

I appreciate the authors effort to attempt a new hybrid metaheuristic algorithm for inverting geoelectrical data

Few things need to be addressed before going for publication

- Many neural networks algorithm works better than other algorithm. What is the significance of using vPSOGWO optimization algorithm.
- What about the computational time and memory for using this algorithm in comparison with other conventional methods?
- Note down the advantages, disadvantages and constraints of the algorithm.
- How principle of equivalence problem can be avoided by using this algorithm?
- Which model of the algorithm works well and give more performance – Forward-Inverse modelling?
- Random weights have been fixed for working out the algorithm. Have the authors applied any specific logic in fixing the weights or else any meaning approach implemented? Clarify.
- What are the types of noises involved in training the algorithm? What about SNR?

---

## Author Response (AR1)

October 25, 2023

To

Prof. Christian Franzke

Handling Editor

Nonlinear processes in geophysics

**Sub:** Reply to Referees comments of manuscript (npg-2022-13)

Dear Sir,

With reference to manuscript (npg-2022-13), first and foremost we would like to thank you for providing the comments raised by two reviewers (Expert referees), which are very beneficial for improving our manuscript and knowledge as well. In view of the comments received from both reviewers, we have replied to reviewer's comments. The reviewers are satisfied with the reply to the comments and recommended for publication of manuscript in its original submitted form. The comments given by both the reviewers are as follows: (i) comments given by first expert (Referee #1) as quoted "**Authors reply to the initial discussions are satisfactory and I recommend the article to proceed for further processing of publication**" and (ii) the comments given second expert (Referee #2) as quoted "**The manuscript is well established and well written. So, I strongly recommend accepting it for publication and do not need any revision**". As the submitted manuscript required no changes, therefore, the same manuscript has been uploaded in "Author's track-changes file". If it require any changes or improvements please let us know and we will be very happy to carry out in future.

We are delighted that this conversation has improved our scientific knowledge. Many thanks to the two reviewers who suggested that our manuscript should be published. As the favourable feedback received from both reviewers, so the manuscript is being forwarded in its original form for processing. We also appreciate NPG for giving the platform for direct discussion with referees.

Thanking you

Yours sincerely

(Upendra K Singh)

Reply to Reviewer #1

First of all we would appreciate your nice and very useful quarries/comment on our submitted manuscript entitled "Inversion, Assessment of Stability and Uncertainty of Geoelectric Sounding data using a New Hybrid Meta-heuristic algorithm and Posterior Probability Density Function Approach" by Kuldeep Sarkar and Upendra K. Singh.

Comment 1: I appreciate the authors effort to attempt a new hybrid metaheuristic algorithm for inverting geoelectrical data

Reply: Dear Reviewer, first of all we would like to thanks for encouraging and appreciating our work to invert the geoelectrical datasets.

Comment 2: Many neural networks algorithm works better than other algorithm. What is the significance of using vPSOGWO optimization algorithm.

Reply: As per my knowledge, there are three main steps in neural networks algorithm: (i) training process, which is time consuming process (ii) validation and (iii) testing. As validation and testing completely depends on how training data is. Second thing is that ANN requires initial guess.

During the training process, ANN uses an optimizer. There are mainly two type of optimizer first is local optimizer. In ANN mostly local optimizer are used (i.e., steepest descent or gradient descent algorithm), therefore most network get stuck at local optima and results become worst, whereas network have least possibility to stuck at local optima in the case of global optimizer (i.e. metaheurastic global optimizer namely PSO, GWO, vPSOGWO etc) and it give global solution.

In contrast to ANN, Global optimization does not required initial guess and there is no such training process. If the initial guess is wrong than it may lead to local minima. For more detail, you may go through given literature: Chen, G. and Yu, J., 2005, August. Particle swarm optimization neural network and its application in soft-sensing modeling. In International Conference on Natural Computation (pp. 610-617). Springer, Berlin, Heidelberg.

Comment 3: What about the computational time and memory for using this algorithm in comparison with other conventional methods?

Reply: We observed using many geophysical examples and found that the computational time for using this algorithm will be higher in comparison with other conventional methods. But this global optimization techniques are computationally inexpensive in terms of both memory requirements and speed. It does not need gradient information, as the gradient-based algorithm does. This allows functions whose gradients are either unavailable or computationally expensive to be solved. You may read the given literature: Chen, G. and Yu, J., 2005, August. Particle swarm optimization neural network and its application in soft-sensing modeling. In International Conference on Natural Computation (pp. 610-617). Springer, Berlin, Heidelberg).

Comment 4: Note down the advantages, disadvantages and constraints of the algorithm.

Reply: **Advantages:** There are many advantages of Global optimization particularly our algorithm (vPSOGWO): (i) any initial guess does not require, (ii) the error between observed and computed data will be lesser than local optimization, (iii) accuracy in model will be remarkably high, (iv) avail to search whole search space (explore) and converge (exploit) to find global minima (Zhang et al., 2021), and (v) it is computationally inexpensive in terms of both memory requirements and speed.

**Disadvantages:** This needs high computational cost in terms of time. But faster in finding result with higher accuracy compared to PSO, GWO and conventional algorithms (Cheng et al., 2021).

**Constrains:** There is no such constraint required in the global optimization techniques.

**References:** Zhang, X., Lin, Q., Mao, W., Liu, S., Dou, Z. and Liu, G., 2021. Hybrid Particle Swarm and Grey Wolf Optimizer and its application to clustering optimization. Applied Soft Computing, 101, p.107061.

Cheng, X., Li, J., Zheng, C., Zhang, J. and Zhao, M., 2021. An Improved PSO-GWO Algorithm with Chaos and Adaptive Inertial Weight for Robot Path Planning. Frontiers in Neurorobotics, 15.

Comment 5: How principle of equivalence problem can be avoided by using this algorithm?

Reply: Using any inversion methods, equivalency problem cannot be avoided. Our findings show that the equivalency problem related to the sedimentary layer has been much minimized

Comment 6: Which model of the algorithm works well and give more performance – Forward Inverse modelling?

Reply: Here there is confusion in Forward and inverse modelling. Using the forward modelling the geophysical data is created from the some specific geological model, whereas geophysical data is inverted by inverse modelling for getting the model i.e. geological model.

In general, the laws of physics provide the means for computing the data values given a model. This is called the "forward problem". In the inverse problem, the aim is to reconstruct the model from a set of geophysical measurements (Snieder, and Trampert, 1999).

Reference: Snieder, R. and Trampert, J., 1999. Inverse problems in geophysics. In Wavefield inversion (pp. 119-190). Springer, Vienna.

Comment 7: Random weights have been fixed for working out the algorithm. Have the authors applied any specific logic in fixing the weights or else any meaning approach implemented? Clarify.

Reply: The vPSOGWO algorithm uses a variable weight which lies between 0 to 1 with iteration, allowing a particular weight at each iteration that help in falling into local minima cause by using constant inertia or linearly decreasing inertia weight (Hu et al., 2018). This weight controls the convergence behavior of vPSOGWO, resulting in reliable solution and faster convergence. Thus this weight is completely different from the weight obtained during the training in neural network. Reference: Hu, Z., Zou, D., Kong, Z. and Shen, X., 2018, June.

A particle swarm optimization algorithm with time varying parameters. In 2018 Chinese Control and Decision Conference (CCDC) (pp. 4555-4561). IEEE.

**Comment 8:** What are the types of noises involved in training the algorithm? What about SNR?

**Reply:** We have applied here metaheurastic algorithm namely vPSOGWO, GWO and PSO to invert the VES data and find the layer parameters. Here, some amount of Gaussian noise is usually added in the synthetic datasets for analysis of the algorithms to make field environment. In this technique training is not required whereas training is one important part of ANN/Machine learning.

**Reply to Referee #2**

Thank you for giving your valuable time to reviewing our manuscript (npg-2022-13). Your thoughtful feedback for the positive reviews and recommendation for publication. We are pleased that this discussion through NPG which has improved our scientific understanding and provided support. Once again, thanks the Referee 2 for strongly recommend accepting it for publication and do not need any revision. We are grateful for your expertise and constructive criticism, which will undoubtedly enhance the quality of our work. Once again, thank you for your invaluable contribution.